# Totipotency of *Daucus carota* L. Somatic Cells Microencapsulated Using Spray Drying Technology

**DOI:** 10.3390/plants10112491

**Published:** 2021-11-18

**Authors:** José Alfredo Santiz-Gómez, Miguel Abud-Archila, Víctor Manuel Ruíz-Valdiviezo, Yazmin Sánchez-Roque, Federico Antonio Gutiérrez-Miceli

**Affiliations:** 1Tecnológico Nacional de México, Instituto Tecnológico de Tuxtla Gutiérrez, División de Estudios de Posgrado e Investigación, Carretera Panamericana Km 1080, Tuxtla Gutiérrez 29050, Chiapas, Mexico; jose.sg@tuxtla.tecnm.mx (J.A.S.-G.); miguel.aa@tuxtla.tecnm.mx (M.A.-A.); victor.rv@tuxtla.tecnm.mx (V.M.R.-V.); 2Dirección de Ingeniería Agroindustrial, Universidad Politécnica de Chiapas, Carretera Tuxtla Gutiérrez-Portillo Zaragoza Km 21+500, Colonia Las Brisas, Suchiapa 29150, Chiapas, Mexico; ysanchez@ia.upchiapas.edu.mx

**Keywords:** carrot, encapsulation, somatic cells, spray drying

## Abstract

The carrot is considered a model system in plant cell culture. Spray drying represents a widely used technology to preserve microorganisms, such as bacteria and yeasts. In germplasm conservation, the most used methods are freeze drying and cryopreservation. Therefore, the aim of this work was to evaluate the effect of spray drying on the viability and totipotency of somatic carrot cells. Leaf, root and stem explants were evaluated to induce callus with 2 mg/L of 2,4-dichlorophenoxyacetic acid (2,4-D). Calli obtained from the stem were cultivated in a liquid medium with 1 mg/L of 2,4-D. Cell suspensions were spray dried with maltodextrin-gum Arabic and maltodextrin-xanthan gum mixtures, two outlet air temperatures (50 and 60 °C) and 120 °C inlet air temperature. Results showed that carrot cells were viable after spray drying, and this viability remained for six months at 8 °C. The totipotency of the microencapsulated cells was proven. Cells that were not spray dried regenerated 24.6 plantlets, while the spray dried cells regenerated 19 plantlets for each gram of rehydrated powder. Thus, spray drying allowed researchers to obtain viable and totipotent cells. This work is the first manuscript that reported the spray drying of plant somatic cells.

## 1. Introduction

The genus *Daucus* (Apiaceae) comprises about 60 species, which are widely distributed and cultivated for their edible tap roots [1] that are generally orange in color, although there are also purple, white, red and yellow varieties. The carrot (*Daucus carota* L.) is a member of the Umbelliferae family, and it is a biennial plant. In addition to being consumed as food, different parts of the carrot can be used for medicinal purposes [2], such as hepatoprotective [3], antisteroidogenic [4], antinociceptive and anti-inflammatory functions [5]. The seeds are used for the treatment of swelling and tumors, and the roots are used as a poultice in mammary and uterine carcinoma as well as for skin cancer [1].

The carrot is considered an efficient model for the study of somatic embryogenesis in higher plants since it is easy to generate morphogenic responses and to synchronize embryogenic cultures [6,7,8]. Several physiological and morphological characteristics that occur during embryogenesis have been described using the carrot as a system [9].

In several cases for germplasm conservation, plant material is preserved by encapsulation [10]. The implementation of encapsulation through the ionic gelling of sodium alginate with calcium chloride provides a technology for the propagation and short-term storage of many plant species [10,11]. Multiple investigations into the encapsulation of somatic embryos in calcium alginate beads using carrots as a model system have been published [6,12,13,14]. The materials and methods of encapsulation of plant material have rarely been varied or modified [12]. These techniques applied to the conservation of plant material can be enriched with new methods, such as spray drying, that are affordable for the availability of plant material, as well as the economic and technological resources in each region of the planet.

Spray drying has been widely used to preserve lactic bacteria [15,16] and eukaryotic microorganisms, such as *Saccharomyces cerevisiae* [17,18]. Spray drying is considered an increasingly important method for obtaining dehydrated lactic bacteria because of its lower cost, shorter processing time and greater availability compared with freeze drying. Similarly, the versatility of spray drying and the progress achieved through the innovation and improvement of techniques have led to greater flexibility in meeting biotechnological requirements, especially low-temperature treatments to avoid a loss of activity at the cellular level [19]. To improve cell survival under heat stress, important strategies must be employed during spray drying, such as the selection and mixture of various encapsulant agents, optimization of parameters, such as inlet and outlet air temperature, and feed flow [20]. Encapsulant agents commonly used include proteins, monosaccharides, disaccharides, polysaccharides and their combinations [21], which leads to the establishment of specific protocols, mainly in bacterial species. In the encapsulation of probiotic bacteria [15,22,23] and yeasts [17,24], it has been reported that maltodextrin in concentrations ranging from 2 to 18% (*w*/*v*) can increase the yield of the product and maintain survival percentages between 80 and 100%. On the other hand, gums have been used mainly in the encapsulation of probiotic bacteria [15] or nitrogen fixers [25] in concentrations ranging from 0.4% to 10% (*w*/*v*) because of their high emulsifying and stabilizing capacity, which makes these encapsulant agents as disaccharides essential for spray drying. 

Biotechnological research, however, based on the use of tissues and plant cells in vitro and encapsulation by spray drying refers to a new perspective in terms of plant propagation and germplasm storage. Therefore, the use of this technology for the encapsulation of plant cells was not reported. Implementing spray drying in plant cells represents an important challenge for plant biotechnology. Thus, the aim of this research was to evaluate the viability and totipotency of somatic carrot cells (*Daucus carota* L.) after spray drying and during storage.

## 2. Results

### 2.1. Plant Material

The seeds showed 100% disinfection using the described protocol. Disinfected seeds of *Daucus carota* L. were placed in MS medium and to germinated after 4 days of incubation. The germination percentage was 83%. These results demonstrate the efficiency of disinfection and germination in carrot seeds. Thus, it was possible to obtain aseptic plantlets to induce callogenesis in stem, leaf and root explants.

### 2.2. Callus Induction

In this study, 2 mg/L of 2,4-D was used for callus induction. In leaf explants, the callus formation was slow compared with stem and root explants. Leaf explants showed 65% callus formation and developed compact, translucent yellow callus, while roots showed 30% callus formation. The callus was compact and yellow-greenish. The stems’ explants showed 100% induction of embryogenic and friable callus and yellow-orange coloration. The calluses from these explants were selected to induce cell suspension.

### 2.3. Cell Suspension Cultures

For cell suspension induction, 1 mg/L of 2,4-D was used. The callus from stem explants showed a suitable response by increasing the concentration of suspension cells. After 28 days, cell concentration was 8.25 × 10^6^ cells/mL, while at 35 days, it was 10.5 × 10^6^ cells/mL. This period of development and growth was considered for harvesting cell pellets. The cell suspension showed 100% viability with multicellular formation, different shapes and clusters after sieving. In general, cell sizes ranged between 5 and 70 µm, but the cells that were possible to encapsulate were clusters with sizes between 5 and 15 µm. Pellets obtained from the cell suspensions were used for spray drying.

### 2.4. Encapsulation by Spray Drying

For spray drying, two mixtures of encapsulant agents with two outlet air temperatures were evaluated. It was observed that the highest yield (73%) of the product was obtained when cells were encapsulated in maltodextrin with gum Arabic at 60 °C outlet air temperature, while the lowest yield (63%) was obtained with the mixture of maltodextrin with xanthan gum at 50 °C outlet air temperature, but the control treatment presented 1.9% of yield (*p* ≤ 0.05) because no encapsulant agents were used. Regarding the water activity (A_W_), it was observed that at a higher outlet air temperature (60 °C) the water activity is reduced. When an outlet air temperature of 50 °C was used, however, the water activity (A_W_) increased both in the maltodextrin with gum Arabic mixture and the maltodextrin with xanthan gum mixture. The treatment with the mixture of maltodextrin with gum Arabic at 50 °C outlet air temperature showed a statistically significant difference compared with other treatments. In the same way, the control treatment presented the highest water activity, and it was statistically significant with respect to the other treatments (*p* ≤ 0.05). The moisture content of powders also depended on the outlet air temperature used. The mixture of maltodextrin with gum Arabic at 60 °C outlet air temperature showed a statistically significant difference (*p* ≤ 0.05) with the lowest moisture percentage of the treatments (2.62), while the mixture of maltodextrin with xanthan gum presented the highest moisture content (3.73). The control treatment presented the highest moisture content, and it was statistically significant with respect to the other treatments (*p* ≤ 0.05). Water activity and moisture content are in sufficient and appropriate ranges to consider the powders with encapsulant agents as stable products compared to the control.

Finally, all evaluated treatments showed 100% cell viability after spray drying. However, in the control treatment, viability was reduced to 34% (*p* ≤ 0.05) because encapsulant agents were not used (Table 1). Therefore, it was possible to show that carrot cells are capable of resisting the spray drying process with an inlet air temperature of 120 °C and 50 °C to 60 °C outlet air temperature (Figure 1).

#### 2.4.1. Stability of Encapsulated Cells during Storage

Water activity (A_W_) was measured for six months of storage at 8 °C to determine the stability of powders obtained in different treatments. Water activity showed increases in each evaluated period. Comparing the results between treatments (Table 2) during the 180 days of storage, the MDGA mixture at 50 °C outlet air temperature showed the highest A_W_ and significant statistical difference (*p* ≤ 0.05) with the other treatments at 30 and 90 days. Moreover, powders obtained with the MDXG mixture at 60 °C outlet air temperature showed the lowest A_W_ values over 120 days, presenting a statistically significant difference (*p* ≤ 0.05) at 30 and 90 days with the MDGA and MDXG mixtures when an outlet air temperature of 50 °C was used in both. When individually evaluating the A_W_ of each treatment with respect to the storage time, it was observed that in each there were increases in the A_W_ in all the storage periods. There was no statistically significant difference between these increases, however. Among the treatments, the type of encapsulant agent and the outlet air temperature used showed a significant effect in the increasing the A_W_ of the products obtained. These parameters allowed selecting the treatment that maintained stability with respect to the water activity.

#### 2.4.2. Viability of Encapsulated Cells during Storage

The viability of cells contained in powders obtained in each treatment was determined from 30 to 180 days of storage at 8 °C (Table 3). All evaluated treatments presented 100% viability during 90 days of storage. Subsequently, from 120 days up to 180 days of storage, the viability was reduced in all treatments, but there was no statistically significant difference. Therefore, the types of encapsulant agents and the outlet air temperatures used did not affect the viability of carrot cells. On the other hand, when each treatment was analyzed individually, it was observed that mixtures MDGA at 50 °C and 60 °C of outlet air temperature and MDXG at 60 °C of outlet air temperature have a significant effect (*p* ≤ 0.05) on the reduction of viability after 150 days of storage. The MDXG mixture at 50 °C outlet air temperature significantly reduced (*p* ≤ 0.05) the viability after 120 days of storage, until reaching the lowest value of viability in all periods (90%) after 180 days of storage (Figure 2).

#### 2.4.3. Cell Totipotency and Redifferentiation

Totipotency and redifferentiation of the encapsulated cells were determined after the cell viability of the products obtained in each treatment had been determined. The rehydrated powder was centrifuged, and the cell pellet obtained was cultured in a semisolid MS medium. Totipotency and redifferentiation were also evaluated in obtained suspension cells that were not used for spray drying (Table 4). Powders with 120 days of storage at 8 °C were used since the viability is reduced after this period. 

Every week, the number of plantlets obtained after cellular redifferentiation was counted. Cell suspensions that were not used for spray drying presented redifferentiated plantlets from the first week to 42 days of culture. The MDGA mixture spray dried at 50 °C of outlet air temperature was the only treatment that showed the redifferentiation of encapsulated cells. In this treatment, the plantlets were redifferentiated after 4 weeks and up to 42 days of culture (Figure 3). At the end of the culture, cells encapsulated by spray drying and the cells that were not encapsulated did not show a significant statistical difference (*p* ≤ 0.05) but did show secondary embryogenesis (Figure 4). 

In the secondary somatic embryogenesis, primary embryos gave rise to secondary embryos in culture. After 14 days of culture of primary embryos, the presence of embryos in the globular stage was observed. Twenty days after culture, embryos were observed in the heart stage, and finally, 30 days after culture, embryos in torpedo and cotyledonary stages were observed. However, after 25 days, it was possible to observe embryos in different stages at the same time in some cultures (Figure 5).

### 2.5. Surface Morphology of Microcapsules

The morphology and surface of the microcapsules and cells without any carrier material is observed in Figure 6. Microcapsules obtained with the mixture MDGA and MDXG at 50 °C outlet air temperature presented spherical shapes with dents and cracks in the surfaces, while microcapsules obtained with the mixture MDGA and MDXG at 60 °C outlet air temperature presented spherical shapes with a dents surfaces free of cracks. Capsules in all treatments ranged in size from 1 to 15 µm. Cells without any carrier material showed an irregular, disorganized and amorphous structure.

## 3. Discussion

Stem explants showed the best induction of embryogenic and friable callus. It has been reported that stem explants are an excellent source for the production of somatic embryos from callus compared with all other explants [2,26]. Our results corroborate that stem explants present efficient callus formation and viable suspension cell cultures. It has also been reported that high concentrations of 2,4-D (above 1 mg/L) added to the culture medium developed different types of individual cells in the suspension cells of *Daucus carota* [2,27]. In this study, suspension cell cultures were established with embryogenic and friable calluses derived from stem explants. In cell suspensions, the formation of individual cells and groups of cells was regularly observed. The cell suspensions showed a rapid proliferation because of the use of friable callus during the first 35 days of culture. The effective induction and proliferation of callus and cell suspensions was due to the use of 2,4-D [2,28]. 

With respect to spray drying (Table 1), the MDGA mixture at 60 °C outlet air temperature presented the highest powder yield (73%), and the MDXG mixture at 50 °C outlet air temperature presented the lowest powder yield (63%). Results showed that the powder yield was statistically affected by the mixture of encapsulant agents and outlet air temperature. These results could be partially explained because the MDGA mixture had a higher solids content (44.0 g of solids/100 mL of solution) compared with the MDXG mixture (30.4 g of solids/100 mL of solution). These concentrations of solids in the encapsulant agent mixtures probably affected the properties of the solutions leading to possible different physicochemical properties of the particles after drying and during storage [29]. In addition, it has been reported that the use of gum Arabic increases the yield of the spray drying [18]. In the control treatment, the yield of the product and the cell viability were statistically different (*p* ≤ 0.05) with all the other treatments, obtaining very low values. In the same way, the values of water activity and moisture content were very high and statistically different (*p* ≤ 0.05) from the other treatments. For carrot cells, encapsulant agents must be used for spray drying because encapsulant agents are not used, the products obtained are highly hygroscopic and unstable with an elastic and viscous texture that is undesirable.

Regarding the water activity, the MDGA mixture at 50 °C outlet air temperature presented the highest water activity, showing a significant statistical difference compared with all treatments. The A_W_ values in all the treatments, however, allowed maintaining cell stability because with these values, the water is less available for biochemical reactions, which allows a longer shelf life of the product. In this sense, Rajam et al. [30] indicated that the A_W_ range of a powder as a product of spray drying must range between 0.1 and 0.5 to keep the microorganisms in a dormant state until before rehydration and cell reactivation. 

On the other hand, the moisture content in the products of the MDGA mixtures at 60 °C outlet air temperatures presented the lowest moisture percentages in the microencapsulation of cells. The moisture at the end of the spray drying was below the 4% required for storage stability [31]. Similarly, in all treatments, moisture within this range was obtained to ensure the stability of the encapsulated cells.

Cell viability after the spray drying in all treatments was 100% and did not show statistical difference (*p* ≤ 0.05) between different treatments. This high viability could be because encapsulant agents protect the cell structure of the microorganism and prevent important damage to DNA, RNA, enzymes and proteins since encapsulant agents are installed in the water-binding sites reflected in a high cell viability [32]. Similarly, this can be explained because drying air transfers latent heat for the evaporation of water from the solution, and because this process occurs quickly (a few seconds), cells were then instantly dried and did not increase their internal temperature. In this case, the concentration of solids did not directly influence cell viability despite the difference in concentrations in the MDGA and MDXG mixtures. Therefore, it could be explained by the relatively high concentration of solids used as encapsulant agents (>30.4 g/100 mL). Some authors reported that concentrations above 20 g/100 mL allow microorganisms to remain viable [33]. 

Regarding the stability of the products during storage (Table 2), the water activity of the treatments was measured in periods of 30 days. The water activity of the products, however, increased during the storage periods and conditions. In this sense, it is important to emphasize that the powders must maintain an A_W_ less than 0.4, and they will be adequately preserved within the range of 0.1 and 0.5 [29], to avoid the proliferation of bacteria and fungi that would reduce the quality and viability of the encapsulated plant cells. After 180 days of storage, the water activity of all the treatments did not exceed the value of 0.3. Thus, the powders maintained stability during the storage period. 

Cell viability during storage was evaluated at different time periods (Table 3) to determine if the encapsulation process of plant cells by spray drying was suitable. Cell viability in all treatments was maintained at 100% during three months of storage. After this period, however, the viability decreased to values ranging from 93 to 90%, but these percentages of viability were still high if we compare the viability of lactic bacteria [20,34] and some yeasts [18] that have been spray dried. The spray drying of somatic carrot cells allows high percentages of viability in a 180-day storage period. To determine that spray drying did not affect the totipotency, carrot cells were analyzed by reactivating cells from the establishment of the pellet in the culture medium to cell redifferentiation. Totipotency was also evaluated in suspensions that were not used for spray drying. This suspension showed the development of plantlets from the first week of culture, while the only treatment that generated plantlets from encapsulated cells was the MDGA mixture at 50 °C outlet air temperature with a storage period of 120 days and drying at 120 °C inlet air temperature (Table 4). Carrot stems and nodal explants have been reported to be appropriate explants for the formation of multiple shoots from embryogenic callus [35]. In the same way, it was reported that despite the bacteriostatic and bactericidal effects, antibiotics can slow, inhibit [36] or stimulate the growth and development of explants [37]. A stimulating effect of cefotaxime has been demonstrated on embryogenesis in wheat and triticale [38]. The activity of cefotaxime can be attributed to its being degraded by plant esterases to produce new metabolites that may have growth-regulating properties [39] or that may mimic plant hormones by having a structure similar to auxins, which act as growth regulators [36,40]. Additionally, cefotaxime could inhibit ethylene production in crops, which is positively correlated with the differentiation of plantlets from callus [41]. The presence of cefotaxime in high concentrations could positively influence somatic embryogenesis, which is reflected in a greater number of plants produced from proembryogenic masses derived from carrot protoplast cultures that are supplemented with 400 to 500 ppm of cefotaxime [42]. Similarly, the application of 500 ppm of cefotaxime to sugarcane callus cultures has been reported to promote somatic embryogenesis and subsequent plant regeneration [43]. Therefore, the use of cefatoxime in the redifferentiation culture medium could have a positive effect on the embryogenic response of the encapsulated cells, proving the presence of somatic embryogenesis in the cultures (Figure 5).

In relation to the morphology of the microcapsules, similar results were reported by Luján-Hidalgo et al. [44] for the encapsulation of a microbial consortium using maltodextrin, gum Arabic, xanthan gum and sodium alginate as coating materials and Loksuwan [45] for the encapsulation of β-carotene with modified tapioca starch, native tapioca starch and maltodextrin as encapsulant agents. This morphology and structure of powders have been reported previously and could be explained by the faster volatilization of the water during drying [46]. Moreover, the formation of dented surfaces of spray dried particles was attributed to the shrinkage of the particles during the drying process [47] and the extensive dented surfaces can probably be attributed to disruptions in carrier material resulting in increased susceptibility to shrinkage during the drying stages [45]. In this first report about the encapsulation of vegetal cells, the encapsulant agents used generated the same type of morphology in the capsules, but the outlet air temperature affected the capsule morphology because at 50 °C outlet air temperature, the capsules presented cracks compared to the capsules obtained at 60 °C outlet air temperature.

Finally, it is evident that in this research, the cells preserved totipotency and also developed secondary somatic embryogenesis (Figure 4). Moreover, it has been shown that spray drying was not lethal to the metabolism of lactic bacteria [48]. Therefore, it is possible to affirm that spray drying is a suitable method for the conservation of cells of *Daucus carota* L. and the MDGA mixture using an inlet air temperature of 120 °C and an outlet air temperature of 50 °C since these conditions allowed preserving cell totipotency for up to four months of storage.

## 4. Materials and Methods

### 4.1. Plant Material

Carrot seeds from the Vita^®^ brand were used, which were purchased at the Rancho Los Molinos company. The seeds were disinfected in 70% ethanol for 5 min, followed by immersion in 5% (*v*/*v*) sodium hypochlorite for 15 min. Afterward, the seeds were washed three times in sterile distilled water [2]. The seeds were placed in tissue culture flasks with ½ MS medium [49] supplemented with 3% sucrose and 2.5% Phytagel^®^. The pH was adjusted to 5.8 ± 0.1. The flasks were incubated for 25 days in a bioclimatic chamber at 25 °C in the dark to evaluate the percentage of disinfection and germination. 

### 4.2. Callus Induction

Plants from disinfected seeds were used to obtain explants that included leaves, stem segments and roots. One-centimeter explants were used and placed in carrot callus induction medium B_5_ [50] supplemented with 3% sucrose and 2 mg/L of 2,4-D and 2.5% Phytagel^®^. The pH was adjusted to 5.8 ± 0.1. They were incubated at 25 °C with a photoperiod of 16 h (approximately 95 to 100 μ mol m^−2^ s^−1^) until embryogenic or friable callus growth was identified. The type of explant was evaluated by the capacity for embryogenic callus formation.

### 4.3. Cell Suspension Cultures

The explant that induced a higher percentage of embryogenic callus formation was selected. Approximately three grams of embryogenic and friable callus formed from this explant were used to induce suspension cultures in Erlenmeyer flasks containing 100 mL of B_5_ liquid medium supplemented with 1 mg/L of 2,4-D and 3% sucrose. The pH was adjusted to 5.8 ± 0.1. Subcultures were performed at 15-day intervals. The cultures were incubated at 25 °C with a 16 h photoperiod (approximately 95 to 100 μ mol m^−2^ s^−1^) at 100 rpm in an orbital shaker.

#### Concentration and Viability of Cells

After five weeks, the suspension cells obtained were sieved under sterile conditions using 200 mesh sieves (WS Tyler, Mentor, OH, USA). The cell viability was determined in broth obtained after sieving according to the exclusion technique of trypan blue. Furthermore, 0.4% trypan blue was used and mixed 1:1 (*v*/*v*) with the cell sample. The reaction was allowed to develop for 1 min, and cells were analyzed under an optical microscope with the aid of a Neubauer chamber. Only cells that showed a blue staining reaction were considered positively stained and not viable, whereas the cells that did not present any coloration were considered viable. Cell viability was calculated using the following equation:(1)$$\mathrm{Cell}\;\mathrm{viability}\;\left(\%\right)=\frac{\mathrm{Number}\;\mathrm{of}\;\mathrm{viable}\;\mathrm{cells}}{\mathrm{Total}\;\mathrm{cell}\;\mathrm{number}}*100$$

For cell concentration, viable cells were counted in the four corner quadrants (A, B, C, D) of the Neubauer chamber and quantified according to the following equation:(2)$$\mathrm{Cell}\;\mathrm{concentration}\;(\frac{\mathrm{Cells}}{\mathrm{mL}})\;=\frac{\left(A+B+C+D\right)\;}{4}{\times 10}^{4\;}\times 2\;$$

Subsequently, 40 mL of cell broth was placed in Falcon tubes; they were then centrifuged at 3500 rpm at 4 °C for 10 min. The cell pellet was reserved for encapsulation by spray drying.

### 4.4. Encapsulation by Spray Drying

To evaluate spray drying, a 2^2^ factorial experimental design was used (Table 5). As encapsulant agents, maltodextrin (Sigma-Aldrich, Saint Louis, MO, USA), gum Arabic (Meyer) and xanthan gum (Meyer) were used for evaluating the effect of two mixtures: maltodextrin-gum Arabic (MDGA) and maltodextrin-xanthan gum (MDXG). The solutions of 30% maltodextrin (*w*/*v*), 14% gum Arabic (*w*/*v*) and 0.4% xanthan gum (*w*/*v*) were prepared separately and hydrated for 24 h in distilled water. The pH was adjusted to 5.8 ± 0.1. Encapsulant agents were mixed in a 1:1 (*v*/*v*) ratio and homogenized with the help of an orbital shaker at 80 rpm for 1 h and sterilized at 15 psi for 15 min. The cell pellet was resuspended in a mixture of encapsulant agents before the spray drying and homogenized in orbital shaking at 100 rpm for 30 min and did not damage the cells. Spray drying was performed in a Buchi Mini Spray Dryer B-290 (Buchi AG, Flawil, Switzerland) with 1 L/h of water evaporation capacity (having a two-fluid nozzle type atomizer) equipped with electronic controls for inlet air temperature, a peristaltic pump for feed control, a cyclone separator for powder collection and a rotameter for air flow. The operating conditions were as follows: the air flow rate (0.58 m^3^/min), an inlet air temperature of 120 °C and two outlet air temperatures of 50 and 60 °C. The feed flow rate for the maltodextrin-gum Arabic mixture at 50 and 60 °C outlet air temperatures was 10 ± 2 mL/min and 5 ± 2 mL/min, respectively, while for the mixture of maltodextrin-xanthan gum at 50 and 60 °C outlet air temperatures was 8 ± 2 mL/min and 3 ± 2 mL/min, respectively. The dryer was fed with sterile distilled water once thermal equilibrium and steady-state conditions were reached. Afterward, the feed was changed over to a cellular solution. The control treatment consisted of a cell suspension without encapsulant agents. The feed flow rate was 0.5 mL/min with the same parameters as the other treatments at 60 °C outlet air temperature because it was not possible to maintain stable conditions at 50 °C outlet air temperature without encapsulant agents. The product obtained was stored at 8 °C in hermetically sealed bags under vacuum. 

#### 4.4.1. Yield, Moisture and Water Activity of Powder

The Yield of the spray drying process was evaluated according to the ratio of grams of powder obtained and the content of total solids in the mixture to be dried according to the equation:(3)$$\mathrm{Yield}\;(\%)=\frac{\mathrm{Powder}\;\mathrm{obtained}\;(g)}{\mathrm{Total}\;\mathrm{solids}\;\mathrm{in}\;\mathrm{the}\;\mathrm{mix}(g)}*100$$

The moisture of the powder obtained as a product of spray drying was determined using the constant weight technique by oven drying. Five grams of the obtained powder was placed in aluminum containers that previously reached the constant weight in an oven at 80 °C. The moisture percentage in each product obtained was determined as indicated in the following equation:(4)$$\mathrm{Moisture}\;\mathrm{content}\;\left(\%\right)=\frac{\mathrm{wet}\;\mathrm{powder}\;\left(g\right)-\;\mathrm{dry}\;\mathrm{powder}\;\left(g\right)}{\mathrm{wet}\;\mathrm{powder}\left(g\right)}\;*100$$

Water activity was determined with the help of a HygroPalm AW1 portable water activity indicator (Rotronic AG, Zurich, Switzerland), placing approximately 3 g of powder in the container until a constant reading was obtained. Water activity was evaluated immediately after spray drying and during each month of storage at 8 °C.

#### 4.4.2. Cell Viability

Cell viability was evaluated by rehydrating 1 g of powder from each treatment in a liquid MS medium supplemented with 200 ppm of cefotaxime [42,51,52] and 400 ppm of Mancozeb^®^ [53,54,55] with orbital shaking at 100 rpm for 24 h. Cell viability was determined according to the trypan blue exclusion technique. Cell viability was evaluated immediately after spray drying and during each month of storage at 8 °C.

#### 4.4.3. Determination of Cell Totipotency and Redifferentiation

Rehydrated powder was centrifuged at 3500 rpm at 4 °C for 10 min. The cell pellet was placed in a solid MS medium without growth regulators and supplemented with 200 ppm of cefotaxime and 400 ppm of Mancozeb^®^. Afterward, it was incubated at 25 °C with a photoperiod of 16 h (approximately 95 to 100 μ mol m^−2^ s^−1^) until the redifferentiation and formation of somatic embryos were identified. These determinations were made immediately after spray drying, as well as during each month of storage at 8 °C. Totipotency and redifferentiation were also evaluated in cells that were not used for spray drying.

### 4.5. Surface Morphology Analysis

The structure and surface of the microcapsules and cells without encapsulant agent was observed using a scanning electron microscope (JEOL, JCM-7000, Tokyo, Japan). Samples were coated with the mixture of gold/palladium by a sputter coating device (DENTON VACCUM, DESK II, Moorestown, NJ, USA). The scanning electron microscope was operated in a high vacuum at 10–15 KV.

### 4.6. Statistical Analysis

A completely random factorial experimental design was used to study the effect of the mixtures and the outlet air temperatures. All the experiments were performed in triplicate. The results were analyzed using an analysis of variance (ANOVA), and the means were compared using the Tukey test (*p ≤* 0.05) with the help of STATGRAPHICS Centurion XVII statistical software.

## 5. Conclusions

This work is the first report on plant somatic cells microencapsulated by spray drying. In this sense, spray drying is a process that allows the viability of stored somatic cells to be maintained for up to six months in refrigeration at 8 °C. Encapsulated cells in MDGA at 50 °C have a development similar to that of nonencapsulated cells after rehydration and reactivation. The encapsulated cells did not lose their totipotency for four months. Results show that, within that period, there was cell redifferentiation and plantlets development via somatic embryogenesis. This research suggests that spray drying could be used as a new method for the conservation of plant material.

## Figures and Tables

**Figure 1 plants-10-02491-f001:**
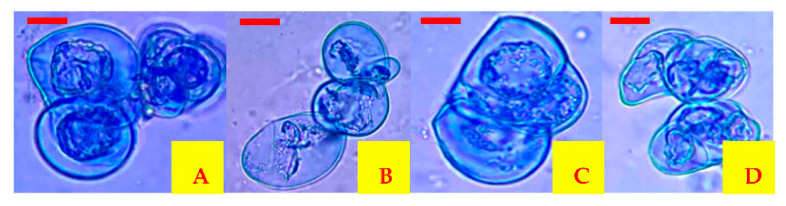
Carrot cells encapsulated by spray drying and reactivated after the process. (**A**) MDGA encapsulated cells at 60 °C outlet air temperature, (**B**) MDGA encapsulated cells at 50 °C outlet air temperature, (**C**) MDXG encapsulated cells at 60 °C outlet air temperature and (**D**) cells encapsulated in MDXG at 50 °C outlet air temperature. Scale bar = 0.005 mm.

**Figure 2 plants-10-02491-f002:**
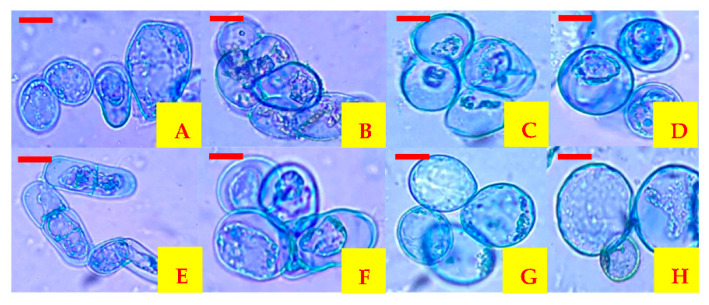
Carrot cells encapsulated by spray drying and reactivated in different storage periods at 8 °C. (**A**) Cells encapsulated in MDGA at 60 °C outlet air temperature after 30 days, (**B**) cells encapsulated in MDGA at 60 °C outlet air temperature after 180 days, (**C**) cells encapsulated in MDGA at 50 °C outlet air temperature after 30 days, (**D**) cells encapsulated in MDGA at 50 °C outlet air temperature after 180 days, (**E**) cells encapsulated in MDXG at 60 °C outlet air temperature after 30 days, (**F**) cells encapsulated in MDXG at 60 °C outlet air temperature after 180 days, (**G**) cells encapsulated in MDXG at 50 °C outlet air temperature after 30 days, (**H**) cells encapsulated in MDXG at 50 °C outlet air temperature after 180 days. Scale bar = 0.005 mm.

**Figure 3 plants-10-02491-f003:**
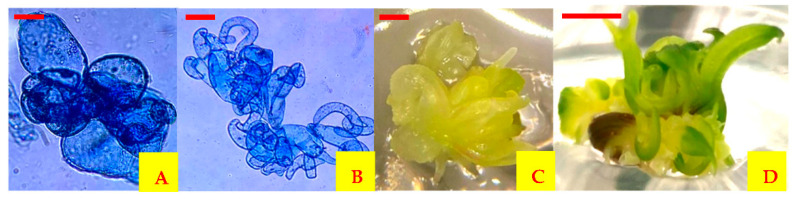
Redifferentiation of encapsulated carrot cells by spray drying. (**A**) Cells encapsulated in MDGA at 50 °C outlet air temperature after 120 days of storage, (**B**) embryogenic cells developed from cells encapsulated in MDGA at 50 °C outlet air temperature grown in MS medium without growth regulators, (**C**) embryos generated from cells encapsulated in MDGA at 50 °C outlet air temperature after 21 days of culture in MS medium without growth regulators, (**D**) plantlets generated from cells encapsulated in MDGA at 50 °C outlet air temperature after 35 days of culture in MS medium without growth regulators. Scale bars = 0.05 mm (**A**,**B**), 1 mm (**C**) and 5 mm (**D**).

**Figure 4 plants-10-02491-f004:**
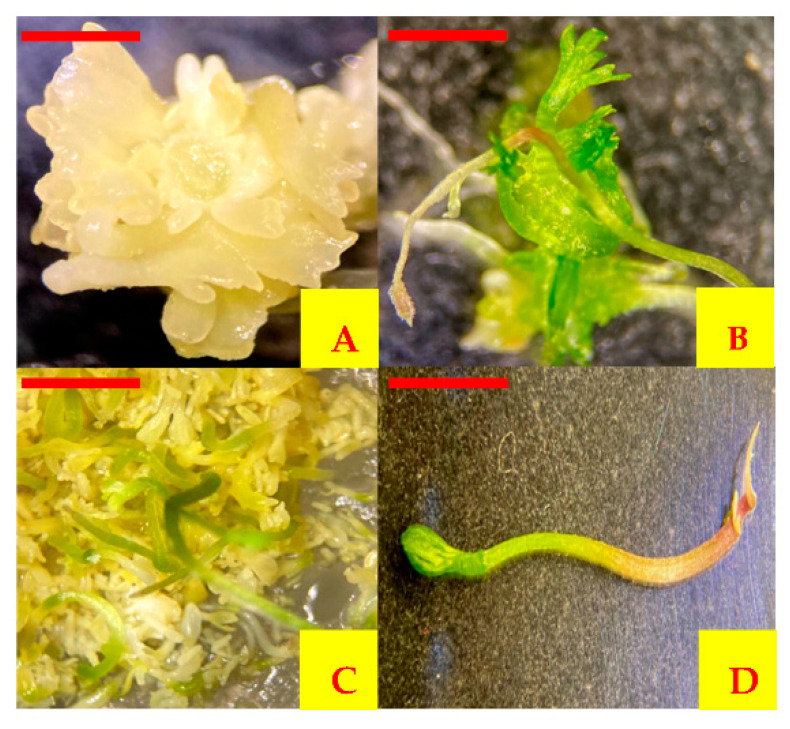
Redifferentiation of carrot cells. (**A**) Somatic embryos generated from cell suspensions without spray drying, (**B**) plantlets regenerated from suspension cells without spray drying, (**C**) secondary somatic embryogenesis induced from encapsulated somatic cells by spray drying in MDGA at 50 °C outlet air temperature, (**D**) plantlets regenerated from encapsulated cells by spray drying. Scale bars = 1 mm (**A**), 5 mm (**B**–**D**).

**Figure 5 plants-10-02491-f005:**
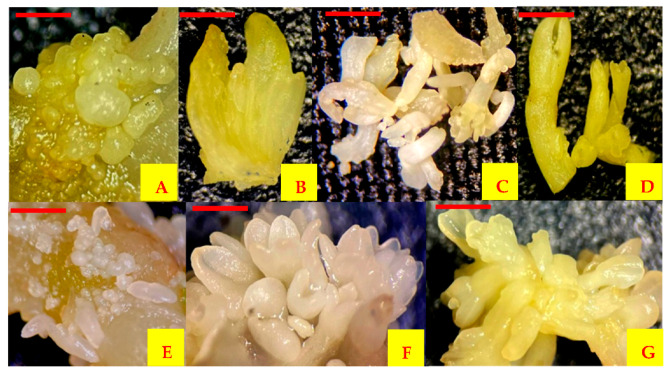
Different stages of somatic embryogenesis of carrot. (**A**) Globular shaped, (**B**) heart shaped, (**C**) torpedo shaped, (**D**) cotyledonary staged, (**E**–**G**) asynchronous culture of embryos with globular, heart and torpedo shapes at the same time. Scale bars = 0.5 mm (**A**,**B**), 1 mm (**C**–**G**).

**Figure 6 plants-10-02491-f006:**
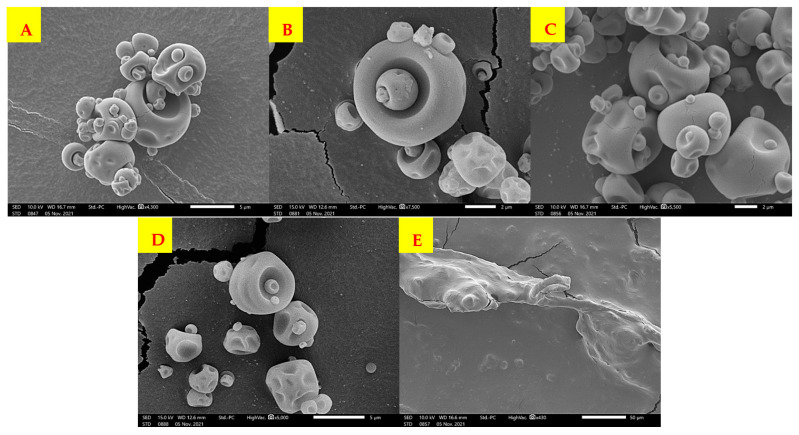
SEM of spray-dried powders of carrot somatic cells. (**A**) MDGA at 50 °C outlet air temperature, (**B**) MDGA at 60 °C outlet air temperature, (**C**) MDXG at 50 °C outlet air temperature, (**D**) MDXG at 60 °C outlet air temperature, (**E**) cells spray dried without encapsulant agents.

**Table 1 plants-10-02491-t001:** Powder yield, water activity, moisture content and cell viability after the spray drying of somatic carrot cells.

Encapsulant Agents	Outlet Air Temperature (°C)	Powder Yield (%)	Water Activity (Aw)	Moisture Content (%)	Cell Viability (%)
Control *	60	1.9c	0.479a	10.54a	34b
MDGA	60	73a	0.119c	2.62c	100a
MDGA	50	70ab	0.186b	3.27bc	100a
MDXG	60	68ab	0.110c	3.68b	100a
MDXG	50	63b	0.142bc	3.73b	100a
LSD		7.46	0.048	1.05	4.88

LSD = Least significant difference (*p* ≤ 0.05), MDGA = maltodextrin-gum Arabic, MDXG = maltodextrin-xanthan gum. Values with the same letter are not significantly different between the treatments. * without encapsulant agents.

**Table 2 plants-10-02491-t002:** Water activity (Aw) of the encapsulated carrot cells during storage at 8 °C.

Encapsulant Agents	OT (°C)	Time of Storage (Days)						
		30	60	90	120	150	180	LSD
MDGA	60	0.120BCa	0.122Ba	0.125Ca	0.128Ba	0.130Ba	0.135Ba	0.025
MDGA	50	0.188Aa	0.192Aa	0.195Aa	0.196Aa	0.201Aa	0.205Aa	0.041
MDXG	60	0.113Ca	0.117Ba	0.121Ca	0.123Ba	0.131Ba	0.143Ba	0.044
MDXG	50	0.148Ba	0.151Aba	0.155Ba	0.160Aba	0.165Aba	0.168Aba	0.048
**LSD**		0.031	0.047	0.024	0.053	0.042	0.053	

LSD = Least significant difference (*p* ≤ 0.05), MDGA = maltodextrin-gum Arabic, MDXG = maltodextrin-xanthan gum, OT = outlet air temperature. Capital letters indicate significant differences between treatments, and lower case letters indicate significant differences between time periods.

**Table 3 plants-10-02491-t003:** Viability of the encapsulated carrot cells stored at 8 °C and reactivated in different periods of time.

Encapsulant Agents	OT (°C)	Time of Storage (Days)						
		30	60	90	120	150	180	LSD
MDGA	60	100Aa	100Aa	100Aa	98Aa	95Ab	92Ac	2.41
MDGA	50	100Aa	100Aa	100Aa	99Aa	96Ab	93Ac	2.14
MDXG	60	100Aa	100Aa	100Aa	97Aab	96Ab	92Ac	2.90
MDXG	50	100Aa	100Aa	100Aa	98Ab	95Ac	90Ad	1.02
**LSD**		0	0	0	3.12	1.802	4.51	

LSD = Least significant difference (*p* ≤ 0.05), MDGA = maltodextrin-gum Arabic, MDXG = maltodextrin-xanthan gum, OT = outlet air temperature. Capital letters indicate significant differences between treatments, and lower case letters indicate significant differences between time periods.

**Table 4 plants-10-02491-t004:** Number of regenerated carrot plantlets per gram of powder with encapsulated cells compared with nonencapsulated cells.

Type of Cell Suspension	Time of Culture (Days)						
	7	14	21	28	35	42	Secondary Embryogenesis
Cells without spray drying	5.3a	10a	14.7a	20.7a	27.6a	24.6a	Yes
Cells with spray drying (OT = 50 °C, MDGA)	0b	0b	0b	3b	13.3b	19a	Yes
LSD	4.03	4.80	4.03	5.39	8.58	13.7	

LSD = Least significant difference (*p* ≤ 0.05), MDGA = maltodextrin-gum Arabic, OT = outlet air temperature. Values with the same letter are not significantly different between the treatments.

**Table 5 plants-10-02491-t005:** Experimental design for the encapsulation of carrot cells by spray drying.

Treatment *	Outlet Air Temperature (°C)	Encapsulant Agent
Control	60	Without encapsulant agents
1	60	Maltodextrin-gum Arabic (MDGA)
2	50	Maltodextrin-gum Arabic (MDGA)
3	50	Maltodextrin-xanthan gum (MDXG)
4	60	Maltodextrin-xanthan gum (MDXG)

* All treatments were carried out in triplicate.

## Data Availability

Not applicable.

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

*faba* L.). BMC Microbiol..

