# Peer review of "Totipotency of Daucus carota L. Somatic Cells Microencapsulated Using Spray Drying Technology"

_plants, 2021, doi:10.3390/plants10112491_

Round 1

Reviewer 1 Report

The authors of the manuscript “Totipotency of Carrot (Daucus carota L.) Somatic Cells Microencapsulated by Spray Drying” reported microencapsulation of carrot cells by spray-drying method. They examined the efficacy of different carrier materials (Maltodextrin-gum Arabic and Maltodextrine-Xanthan gum) and outlet air temperature on microencapsulation of carrot cells by spray drying. Also, the viability and totipotency of microencapsulated carrot cells were evaluated. The authors should describe in the introduction why they chose Maltodextrin-gum Arabic and Maltodextrine-Xanthan gum for microencapsulation of carrot cells. Surface morphological analysis of spray-dried carrot cells powder should be carried out using scanning electron microscopy to confirm the normal cells present in the dried powder.

Some specific comments:

L15: ‘Calli obtained from the stem were’ instead of ‘Callus stem were’.

L32: Delete ‘(Daucus carota L.)’

L37: germplasm instead of germoplasm.

L37-38: Citation is required.

L56: temperatura?

L70: Germination percentage was influenced by several factors. Please do not compare this result with others. Also, this is not a new finding in well-studied species.

L75: 2.2 Callus induction – what about leaf and root explants.

L102: (3.73%)? Correct the value according to Table 1.

L109: Table 1. The control is missing. For control, spray drying of carrot cells without carrier materials.

 L197: plantlets instead of seedlings. Correct it throughout the text.

L197: cellular or embryogenic redifferentiation? Please be specific adventitious regeneration or somatic embryogenesis. Also, indicate the number of days required for various stages of somatic embryo development.

L216: Figures 3C and D introduce the scale bar.

L236: Figure 4 introduces scale bar.

L242: Disinfected seeds of Daucus carota L. were placed in MS medium and germinated after 4 days of incubation. Move it to results.

L247: Please indicate the concentrations of 2,4-D.

L254-257: Move it to results.

L340: 5.8±1? ±0.1

L343: 4.3 Cells suspensión cultures – indicate the amount of calli used for initiation cell suspension.

L379-381: Please indicate the other operating conditions.

L412: 200 ppm of cefotaxime and 400 ppm of mancozeb. Cite relevant articles or explain why they chose these compounds to eliminate contaminants.

L422: formation of embryogenic structures were identified. Please present the data.

The authors should include Scanning Electron Microscopy photographs of and to study the impact of spray drying on normal cells without matrix, effects of antibiotics on somatic embryo development and conservation.

Reviewer 2 Report

Dear Authors,

I have found your work " Totipotency of Carrot (Daucus carota L.) Somatic Cells Microencapsulated by Spray Drying" very interesting. You have done extensive work of  importance in my opinion and I think your obtain results and conclusions could interest many researchers and readers. You presented an innovative approach and there are fine well- documented observations, but I think that You should take a count modification of this article. I recommend publishing it in "Plants" after correcting listed below suggestions:

-The general problem is that the text is written in specific English, the Authors use expressLineion coming from Spanish (for example: Line 56: “temperature” instead of “temperature”, line 79: “suspensión” etc. Therefore, in my opinion, in that form manuscript is not suitable for publication. The text should be thoroughly improved; both at the linguistic and stylistic levels.

- Taking into account the complicated methodology of work, a good solution that would facilitate its understanding would be to develop an appropriate scheme

Title:

“Totipotency of Carrot (Daucus carota L.) Somatic Cells Microencapsulated by Spray Drying” -  I suggest modifying it as follows: ”Totipotency of Daucus carota L. Somatic Cells Microencapsulated using Spray Drying technology/method” because, as I suppose the "spray drying" process is the method which prepares the micropropagules for encapsulation.

Abstract:

This part of an article in not written according to the instruction for Authors of "Plants", there are not background, it should be rewritten according to the rules.

Keywords:

Daucus carota L., somatic cells, encapsulation, spray drying – carrot, encapsulation, somatic cells, spray drying

Introduction:

Line 31: …medicinal purposes [2] – it would be good to expand this example

Line 32: Carrot (Daucus carrota L.) … - D. carrota

Line 64: Implementing spray drying in plant cells represents an important challenge for plant biotechnology. – I am still not convinced why the Authors chose this method, please prove it

Results:

Figure 1, 2, 3, 4 – there is the lack of bar scale 

With best regards,

Reviewer 3 Report

Hello, 

I would like to say taht the design of the work is made very well, statisticaly for in vitro culture "triplicate" is enough, the methods are choosen good. The explanation why spray drying can be used is emphasized and the culture conditions are described properly. 

I do not find myself as an expert for in vitro culture, so I was observing the article from the point of in vitro expert.

Article is scientificaly interesting.

Regards,

Zvjezdana

Round 2

Reviewer 1 Report

L109: Table 1. The control is missing. For control, spray drying of carrot cells without carrier materials.

Due to the type of dryer used, it would be necessary to feed high volumes of cell suspension to obtain a considerable amount of powder as a product. The pellets used in this research were obtained in Erlenmeyer flask and considered the period of cell growth of the carrot, as well as the treatments using two outlet air temperatures A control treatment was not used since only cell suspensions had to be dried at 50°C and 60°C as controls, but the mass of powder as a product would be minimal. For this reason, in most research where spray drying is used, there is no direct control (Huang et al., 2017), but in particular cases the effect of encapsulating agents is compared with lyophilized cells without any encapsulating material (Chandralekha et al., 2016). Furthermore, lyophilized cells are considered as controls when new encapsulating materials are evaluated. In our case, we did not evaluate new encapsulating materials, and, because of the above, we consider that a control treatment is not necessary.

Chandralekha, A., Tavanandi, A. H., Amrutha, N., Hebbar, H. U., Raghavarao, K. S. M. S. & Gadre, R. (2016). Encapsulation of yeast (Saccharomyces cereviciae) by spray drying for extension of shelf life. Drying Technology, 34(11), 1307-1318. https://doi.org/10.1080/07373937.2015.1112808

Huang, S., Vignolles, M.L., Chen, X. D., Le Loir, Y., Jan, G., Schuck, P. & Jeantet, R. (2017). Spray drying of probiotics and other food-grade bacteria: A review. Trends in Food Science & Technology, 63, 1-17. https://doi.org/10.1016/j.tifs.2017.02.007

Response to authors: This is the first study in plants; thus, control is needed.

L422: formation of embryogenic structures were identified. Please present the data.

Done. Data or evidence of embryo formation is presented in Figure 3.

Response to authors: Please present the data. Embryo formation (evidence) is not an issue in the studied species.

L197: cellular or embryogenic redifferentiation? Please be specific adventitious regeneration or somatic embryogenesis. Also, indicate the number of days required for various stages of somatic embryo development.

Done. Specifically, it is cellular redifferentiation since in the process cells are encapsulated, and during reactivation the cells are released from the microcapsules and can be redifferentiated in the culture medium. In this research the cells were redirected by somatic embryogenesis as shown in Figure 3.

Response to authors: Please present the somatic embryogenesis data.

The authors should include Scanning Electron Microscopy photographs of and to study the impact of spray drying on normal cells without matrix.

Due to the time determined by the editor for sending the corrections of the manuscript, it was not possible to take the photographs, coupled with the fact that the equipment was not available in our state. In this sense, we consider that the information that would be obtained with the photographs with scanning electron microscopy is not much compared to what is presented in the manuscript and we do not consider it important to carry out for the purpose of a better discussion of results. So if it was necessary to add these photographs we would have to send the samples to others laboratories of our country to take the photos but we do not know the waiting time for the processing of the samples and the delivery of the photographs.

Response to authors: Please do the SEM analysis and present the results. It is very important evidence for this first study. 

Round 3

Reviewer 1 Report

The authors addressed most of the comments.